# Different Tectonic Evolution of Fast Cooling Ophiolite Mantles Recorded by Olivine-Spinel Geothermometry: Case Studies from Iballe (Albania) and Nea Roda (Greece)

**Micol Bussolesi** [1], **Giovanni Grieco** [2,*], **Alessandro Cavallo** [1] and **Federica Zaccarini** [3]

[1] Department of Earth and Environmental Sciences—DISAT, University of Milan-Bicocca, 20126 Milan, Italy; micol.bussolesi@gmail.com (M.B.); alessandro.cavallo@unimib.it (A.C.)
[2] Department of Earth Sciences, University of Milan, 20133 Milan, Italy
[3] Department of Applied Geological Sciences and Geophysics, University of Leoben, A-8700 Leoben, Austria; federica.zaccarini@unileoben.ac.at
[*] Correspondence: giovanni.grieco@unimi.it

**Abstract:** Mg-Fe$^{2+}$ diffusion patterns in olivine and chromite are useful tools for the study of the thermal history of ultramafic massifs. In the present contribution, we applied the exponential modeling of diffusion patterns to geothermometry and geospeedometry of chromitite ores from two different ophiolite contexts. The Iballe ophiolite (Northern Albania) hosts several chromitite pods within dunites. Primary and re-equilibrated Mg#, estimated by using an exponential function, provided re-equilibration and primary temperatures ranging between 677 and 996 °C for chromitites and between 527 and 806 °C for dunites. Cooling rates for chromitites are higher than for dunites, suggesting a different genesis for the two lithologies, confirmed also by spinel mineral chemistry. Chromitites with MORB affinity formed in a SSZ setting at a proto-forearc early stage, explaining the higher cooling rates, while dunites, with boninitic affinity, were formed deeper in the mantle in a more mature subduction setting. At the Nea Roda ophiolite (Northern Greece) olivine in chromitites do not show Mg-Fe variations, and transformation into ferrian chromite produced "fake" diffusion patterns within chromite. The absence of diffusion patterns and the low estimated temperatures (550–656 °C) suggest that Nea Roda chromitites were completely re-equilibrated during an amphibolite-facies metamorphic event that obliterated all primary features.

**Keywords:** olivine-spinel equilibrium; geothermometry; diffusivity patterns; cooling rates; ophiolites; Iballe; Nea Roda

## 1. Introduction

Spinels crystallize in a wide range of solid solutions at different PT conditions. They are refractory and resistant to alteration, and in the case of chromite, among the first phases to crystallize [1–3]. Due to their properties, they are used as petrogenetic indicators [2,4,5], as their mineral chemistry is a function of parental melt composition and geodynamic setting, as well as temperature and pressure.

The crystal structure of the spinel allows a certain number of cation substitutions both at the tetrahedral and at the octahedral sites. The Mg-Fe$^{2+}$ exchange between spinel and olivine is of particular importance, as it is temperature-dependent [2]. The chemical composition of olivine and spinel, linked to the equilibrium constant of the exchange reaction (Kd) allowed the calibration of several geothermometers [4–8]. The olivine-spinel geothermometers potentially work over a wide range of temperatures, from high-T conditions down to at least 650 °C, until the cessation of the elemental exchange [8–10].

One of the factors that affect the possibility to reach equilibrium is the "*r*" factor [olivine/(olivine + chromite)], first mentioned by Engi [7] and later studied by Grieco et al. [9]. Chromitites and their associated dunites show a wide range of r values, from high (close

to 1) in dunites with accessory spinel, to the lowest values in massive chromitites (close to 0), where there is spinel predominance over olivine. Given the high variability of chromitite textures, there is a wide range of intermediate values dependent on the texturally controlled volume proportions of olivine vs. spinel. During re-equilibration at high r values (peridotites with accessory spinel), olivine composition is almost unaffected, whereas spinel can change its composition considerably. On the contrary, where the re-equilibration occurs at low r values, it is the chromite composition that is relatively unaffected by olivine-spinel exchange [9].

Olivine-spinel exchange results in regular trends of Mg# [$Mg/(Mg + Fe^{2+})$] close to crystal rims, known as diffusivity patterns. As the highest Mg# variation occurs in the few micrometers closest to the rim, modeling the diffusivity curves with an exponential function proved to be the most reliable method to infer primary and re-equilibrated Mg# values [11]. These values can in turn be used to estimate primary and re-equilibrated temperatures through geothermometry.

The present contribution aims to provide new insights into the evolution of two partially serpentinized chromitites hosted in the ophiolites of Iballe (Albania) and Nea Roda (Greece) through the study of olivine-spinel diffusion profiles.

## 2. Geological Setting

### 2.1. Iballe

Albanian ophiolites stretch from North to South across the whole country. According to geological reconstructions, these bodies are remnants of oceanic lithosphere derived from the Mesozoic Pindos-Mirdita basin, developed as a seaway between Apulia and Pelagonia microcontinents [12,13]. Iballe chromitites are hosted within a well-exposed ophiolite sequence cropping out in the northern part of the country, within the Mirdita ophiolite.

The Albanian Alps (referred to as Dinarides or Albanides) are characterized by continental and oceanic tectonic units that attest to the evolution of the Jurassic Tethys Ocean [12,14]. Their geodynamic evolution began with the break-up of Gondwana during Middle-Late Triassic, leading to the development of the Jurassic Tethys Ocean. From Middle Jurassic to Early Cretaceous, a new geodynamic regime led to Tethys subduction and successive obduction of oceanic crust fragments onto continental margins. Ophiolites represent the suture zones formed during the closure of the oceanic basin [12].

Based on petrological and geochemical data, the entire Mirdita ophiolite is composed of several ophiolitic massifs divided into two different belts: western and eastern-type ophiolites [15,16] (Figure 1). The western ophiolites (Krabbi, Gomsique and Puka massifs) consist of clinopyroxene-bearing harzburgites with rare chromitite occurrences, plagioclase lherzolites, rare dunites and amphibole peridotites. The volcanic sequence is mainly represented by MOR-type basalts [17]. The eastern ophiolites (Tropoje and Kukes massifs) consist of depleted harzburgites and dunites with abundant chromitite bodies, covered by a pyroxenite layer, a plutonic sequence (gabbronorite, gabbro, diorite, plagiogranite) and a sheeted dyke complex [17]. Iballe chromitites belong to the Krabbi massif, one of the western ophiolites [13], and is mainly constituted by lherzolites and minor plagioclase-bearing peridotite [15] (Figure 1). However, some authors report the location of the Krabbi massif within the eastern-type ophiolites [18], as the limit between the two types is yet not well defined. The genesis of the two ophiolite belts is debated. Some models argue for the formation of the western ophiolites by seafloor spreading and of the eastern ophiolite by intraoceanic subduction [19,20]. Bortolotti et al. [21] argued for a formation in the same oceanic basin which records a first MOR spreading center event subsequently evolving into a supra-subduction setting. The same authors propose a formation of eastern ophiolites from ascending mantle diapirs in an incipient arc, whereas the western ophiolites are formed in a supra-subduction setting. Other authors, however, argue that all Albanian ophiolites were formed in a supra-subduction setting [15].

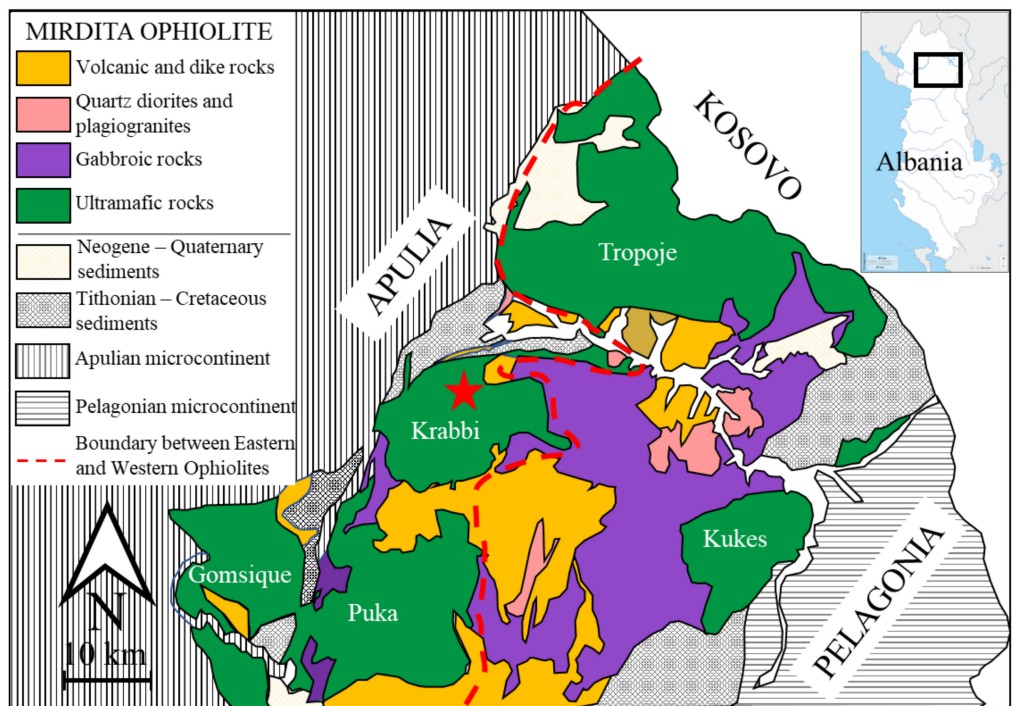

**Figure 1.** Simplified geological map of the Mirdita ophiolite (coloured). The red star represents the sampling location. Modified after Dilek et al. [15].

Albanian ophiolites are characterized by a high metallogenic potential [22], with important Cr, Cu, Ni and Fe mineralization. Olivine and magnesite are also potentially interesting raw materials. All major deposits in the Mirdita ophiolite are hosted by the harzburgitic-dunitic mantle, while low potential chromite deposits are hosted by lherzolite sections of the belt.

### 2.2. Nea Roda

Nea Roda is geographically located in the Chalkidiki Peninsula, Northern Greece, ~80 km SE of the city of Thessaloniki. The Chalkidiki Peninsula is a geologically complex region, which comprises several geotectonic zones belonging to the Hellenides belt, tectonically emplaced during the Mesozoic to Cenozoic evolution of subduction and collision. The Hellenides are part of the Alpine-Himalayan orogenic belt and were formed by the collision of African and Euroasiatic plates in the late Cretaceous. They are separated into two tectonic groups: the External Hellenides and the Internal Hellenides. The latter are further divided into tectonic zones. The main ones are, from west to east: Pelagonian Zone (PZ), Vardar Zone, Serbo-Macedonian Massif (SMM) and Rhodope Massif (RM), all roughly oriented NNW-SSE [23].

The Nea Roda ophiolite occurs in the SMM, a metamorphic massif constituting the crystalline basement of the Alpine orogenic belt in the Balkan Peninsula [24]. The massif comprises amphibolite-facies metamorphic rocks intruded by Late Cretaceous to Miocene granitoids and locally intercalated with basic-ultrabasic units [24]. The SMM is divided into a lower unit, the Kerdillyon Unit, and an upper one, the Vertiskos Unit, separated by a SW-dipping thrust [25]. The Kerdyllion unit, cropping out in the eastern part of the SMM, is composed of migmatized gneiss and schist intruded by Triassic to Cenozoic granitoid rocks [26]. The Vertiskos unit, cropping out in the central part of the SMM, consists of an alternation of gneiss and schist hosting mafic-ultramafic bodies [27].

The Nea Roda massif, cropping out in the SMM, is an ultramafic body thrust onto the Vertiskos Unit (Figure 2) [28]. It contains magnesite bodies of hydrothermal origin, associated with dunites [28,29], rimmed to the East by amphibolites. Small chromitite lenses with textures ranging from massive to disseminated are hosted within dunite dykes enveloped within massive harzburgites [28].

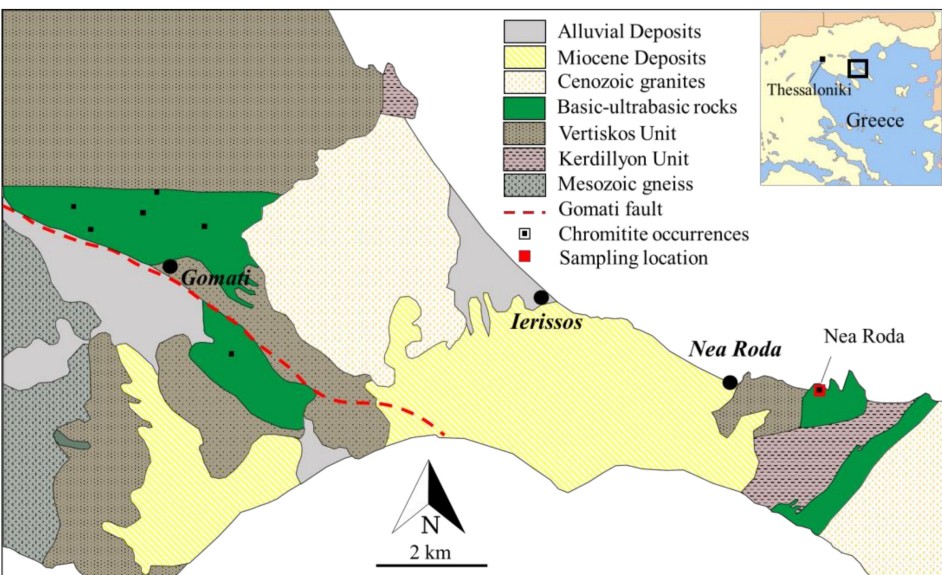

**Figure 2.** Modified excerpt of the 1:50,000 scale geological map of the Ierissos sheet [30]. The red square indicates the sampling location.

## 3. Materials and Methods

Chromitite and dunite samples were collected from the Iballe mine, in Northern Albania (Figure 3a), and from the Nea Roda ophiolite in the Chalkidiki peninsula, Northern Greece (Figure 3b).

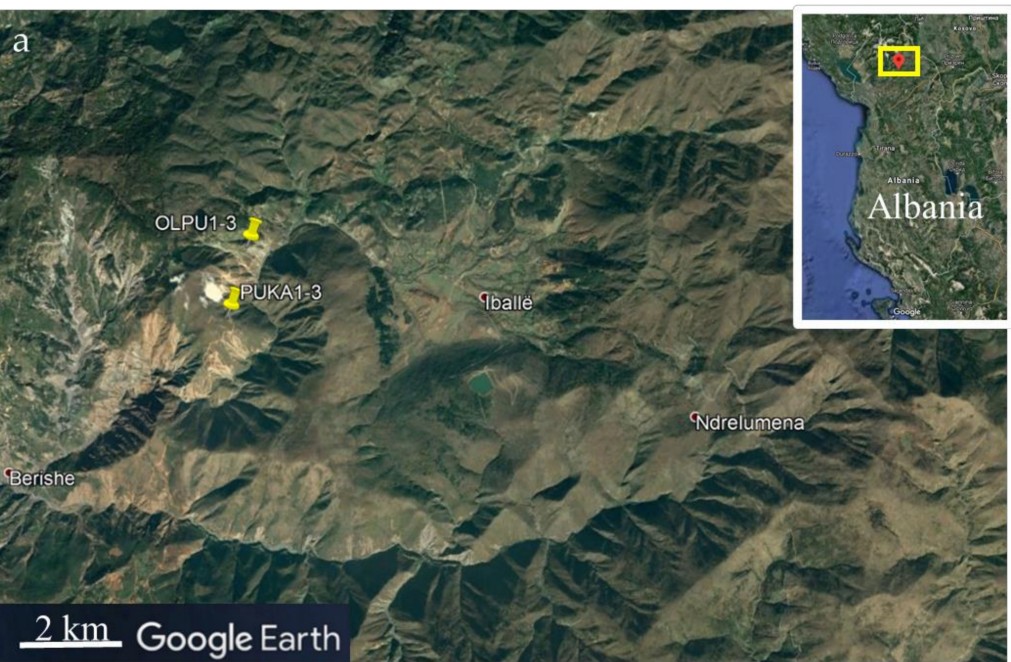

**Figure 3.** *Cont.*

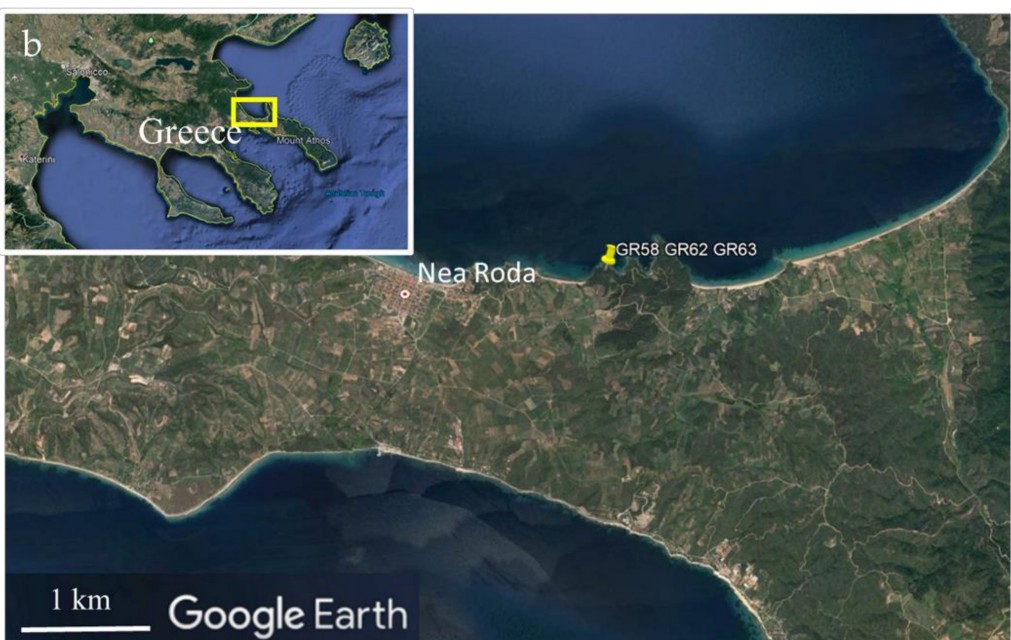

**Figure 3.** Sampling position (Google Earth) of (**a**) Iballe chromitite (42°11′5.13″ N, 19°57′17.03″ E) and dunite samples (42°11′29.57″ N, 19°57′24.34″ E); (**b**) Nea Roda nodular and banded chromitites (40°22′49.56″ N, 23°57′8.49″ E).

Chromitite bodies occur at Iballe in small lenses. Massive chromitite samples were collected from the stock of the working mine (Figure 4a), and fresh dunite samples were collected from outcrops (Figure 4b).

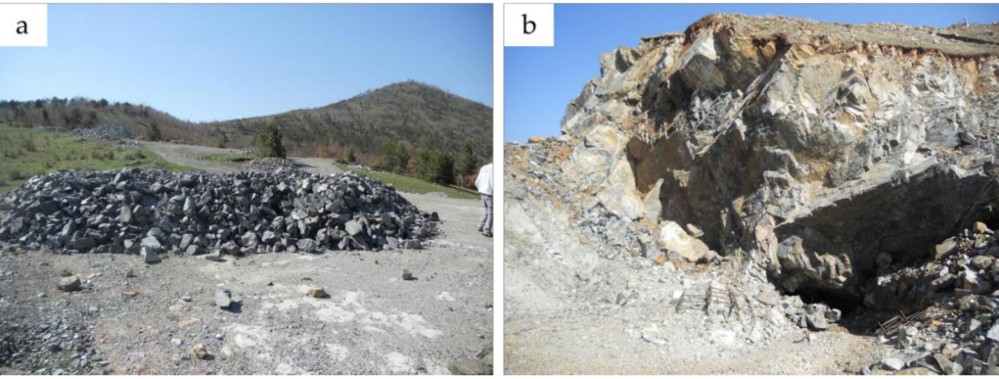

**Figure 4.** Pictures of (**a**) Stock of chromitite ore outside of Iballe mine; (**b**) fresh dunite outcrop at Iballe.

Small chromitite bodies occur to the East of the Nea Roda village. Samples were collected from a peridotite outcrop on the coast, where rare dunite dykes are enclosed within host harzburgite (Figure 5a). Dunites are mostly fresh, and host small chromitite lenses up to 10 cm thick (Figure 5b). The mineralized layers are marked by nodular and banded chromitites with up to 20% chromite content.

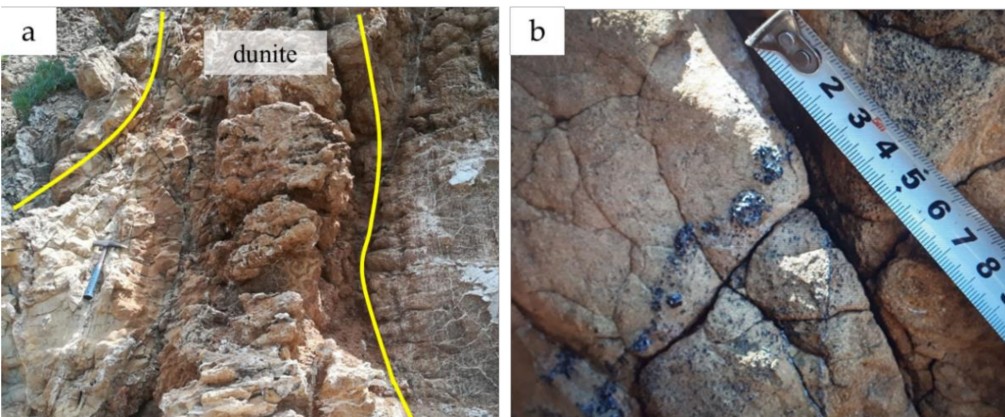

**Figure 5.** Pictures of (**a**) dunite dyke hosted within harzburgite at Nea Roda; (**b**) small chromitite bands within dunite dykes.

The composition of chromite and olivine was determined through a JEOL 8200 electron microprobe equipped with a wavelength dispersive system (SEM-WDS) at the Earth Sciences Department (University of Milan). The system was operated using an accelerating voltage of 15 kV, a sample current on brass of 15 nA, a counting time of 20 s on the peaks and 10 s on the background. A series of natural minerals was used as standards: wollastonite for Si, forsterite for Mg, ilmenite for Ti, fayalite for Fe, anorthite for Al and Ca, chromite for Cr, nickeline for Ni, rhodonite for Mn and Zn, and metallic V for that element. The approximate detection limit is 0.01 wt% for each element. $Fe^{3+}$ was recalculated from microprobe analyses assuming perfect stoichiometry, based on 8-oxygen formula.

Part of the samples was analyzed through a JEOL 8200 electron microprobe at the University of Leoben, Austria. The system was operated with an accelerating voltage of 15 kV and beam current of 10 nA, a counting time of 20s on the peaks and 10s on the background. The elements were analyzed using the K$\alpha$ line. Specimens of chromite, rhodonite, ilmenite, albite, pentlandite, wollastonite, kaersutite, sphalerite and metallic vanadium were used as standards. The following diffracting crystals were used: TAP for Na, Mg and Al; PETJ for K, Si and Ca; LIFH for Ti, V, Cr, Zn, Mn, Fe and Ni.

Analyses have been arranged in grids and traverses along olivine-spinel couples, with a higher density of point analysis close to the olivine-spinel boundaries. Diffusivity profile modeling utilized OriginPro software (version OriginPro 8, OriginLab Corporation, Northampton, MA, USA). Temperature recalculations followed the methodology of Ballhaus et al. [6] and Fabriès [8] calibrations (the latter was used for the cooling rate evaluation).

## 4. Results

### 4.1. Mineralogy and Texture

#### 4.1.1. Iballe

Chromitites and dunites are mostly fresh, with little to absent serpentinization. Chromitites show a massive texture, with submillimetric to millimetric subhedral chromite crystals rarely altered to ferrian chromite. Olivine, the primary silicate associated with chromite, is an interstitial phase. Serpentine is not widespread, but where present, partially replaces olivine grains along the rim (Figure 6a).

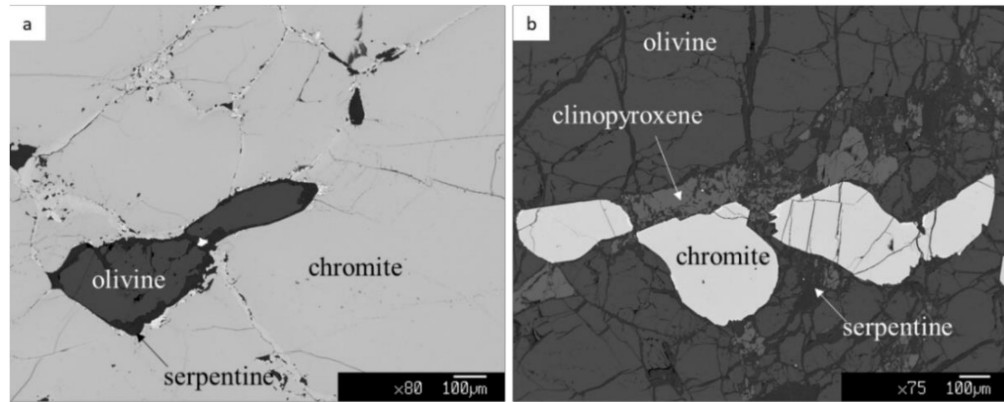

**Figure 6.** BSE images of Iballe (**a**) massive chromitite; (**b**) fresh dunite.

Dunites present a mineralogical assemblage of olivine, minor clinopyroxene and dispersed, submillimetric, chromite. Olivine crystals are fractured and partially replaced by serpentine along the rims. Chromite has a polygonal shape with rounded edges, and is partially fractured. No ferrian chromite alteration was detected (Figure 6b).

### 4.1.2. Nea Roda

Chromite crystals within dunite-hosted chromitites have polygonal to irregular shapes. The grain size ranges from submillimetric to millimetric (Figure 7a,b), and chromite has been partially altered to ferrian chromite (Figure 7a). The silicate gangue consists of olivine grains partially replaced by serpentine, forming a typical mesh texture. Rare chlorite crystals were detected close to ferrian chromite alteration.

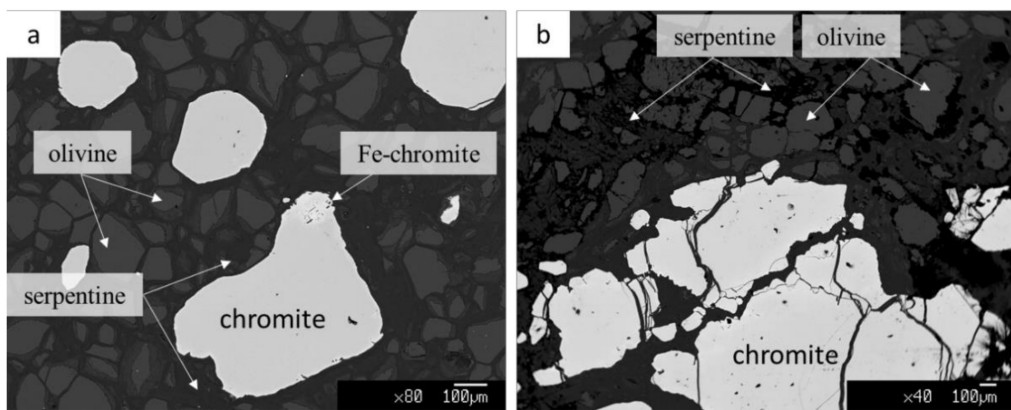

**Figure 7.** BSE images of Nea Roda chromitites showing alteration to ferrian chromite and serpentinization; details of: (**a**) banded chromitite; (**b**) nodular chromitite.

### 4.2. Mineral Chemistry

#### 4.2.1. Iballe

Chromite average compositions (Table 1) are based on core analyses of fresh chromitite and dunite samples. Complete results of electron-microprobe analyses used for the assessment of Mg# variability from core to rim are reported in the Supplementary Materials. Analyzed samples consist of dunites with dispersed spinel (OLPU-) and massive chromitites (PUKA-).

**Table 1.** Major and minor elements average composition and standard deviation of chromite cores from Iballe dunites and chromitites; Mg# = [Mg/(Mg + Fe$^{2+}$)]; Cr# = [Cr/(Cr + Al)].

| Lithology | Dunite | | | | Massive Chromitite | | | |
|---|---|---|---|---|---|---|---|---|
| Sample | OLPU-1 | | OLPU-3 | | PUKA2 | | PUKA3 | |
| wt% | Avg | St.Dv. | Avg | St.Dv. | Avg | St.Dv. | Avg | St.Dv. |
| TiO$_2$ | 0.06 | 0.04 | 0.08 | 0.02 | 0.12 | 0.02 | 0.06 | 0.03 |
| Al$_2$O$_3$ | 8.88 | 0.26 | 8.11 | 0.27 | 26.39 | 0.16 | 24.06 | 0.64 |
| Cr$_2$O$_3$ | 61.92 | 0.40 | 63.00 | 0.52 | 43.35 | 0.31 | 46.07 | 0.39 |
| V$_2$O$_3$ | 0.17 | 0.02 | 0.00 | 0.00 | 0.00 | 0.00 | 0.20 | 0.04 |
| Fe$_2$O$_3$ | 1.90 | 0.28 | 0.06 | 0.09 | 3.36 | 0.28 | 2.90 | 0.49 |
| FeO | 16.99 | 0.89 | 20.24 | 0.71 | 12.20 | 0.27 | 11.73 | 0.24 |
| MgO | 10.84 | 0.66 | 8.36 | 0.38 | 15.93 | 0.20 | 15.84 | 0.17 |
| tot | 101.19 | 0.42 | 100.09 | 0.63 | 101.61 | 0.57 | 101.19 | 0.48 |
| Ti | 0.00 | 0.00 | 0.00 | 0.00 | 0.00 | 0.00 | 0.00 | 0.00 |
| Al | 0.34 | 0.01 | 0.32 | 0.01 | 0.91 | 0.01 | 0.84 | 0.02 |
| Cr | 1.60 | 0.01 | 1.68 | 0.01 | 1.01 | 0.01 | 1.08 | 0.01 |
| V | 0.00 | 0.00 | 0.00 | 0.00 | 0.00 | 0.00 | 0.00 | 0.00 |
| Fe$^{3+}$ | 0.05 | 0.01 | 0.00 | 0.00 | 0.07 | 0.01 | 0.06 | 0.01 |
| Fe$^{2+}$ | 0.46 | 0.03 | 0.57 | 0.02 | 0.30 | 0.01 | 0.29 | 0.01 |
| Mg | 0.53 | 0.03 | 0.42 | 0.02 | 0.70 | 0.01 | 0.70 | 0.01 |
| Cr# | 0.824 | 0.004 | 0.839 | 0.005 | 0.524 | 0.002 | 0.562 | 0.008 |
| Mg# | 0.532 | 0.028 | 0.424 | 0.019 | 0.699 | 0.007 | 0.706 | 0.006 |

The MgO content of dunite-hosted chromite ranges between 7.18 and 11.54 wt%, the FeO content between 16.10 and 22.50 wt% and the Fe$_2$O$_3$ between 0.02 and 2.46 wt%. The Cr$_2$O$_3$ and the Al$_2$O$_3$ contents vary between 61.33 and 63.85 wt% and between 7.23 and 9.22 wt%, respectively. Mg# ranges between 0.36 and 0.56, and Cr# = [Cr/(Cr + Al)] ranges between 0.82 and 0.86.

The MgO content of chromitite spinels range between 15.52 and 16.38 wt%, the FeO content range between 11.25 and 12.66 wt% and the Fe$_2$O$_3$ between 1.67 and 3.90 wt%. The Cr$_2$O$_3$ content is lower than in dunite spinels, and range between 42.87 and 46.76 wt%, while the Al$_2$O$_3$ content is higher, ranging from 22.64 to 26.69 wt%. These compositional differences result in a higher Mg# (0.69–0.72) and lower Cr# (0.52–0.58) with respect to dispersed spinels in dunites.

Olivine average compositions and standard deviations of dunites and massive chromitites are reported in Table 2.

**Table 2.** Major and minor elements average composition and standard deviation of olivine cores from Iballe dunites and chromitites; Mg# = [Mg/(Mg + Fe$^{2+}$)].

| Lithology | Dunite | | | | Massive Chromitite | | | |
|---|---|---|---|---|---|---|---|---|
| Sample | OLPU-1 | | OLPU-3 | | PUKA2 | | PUKA3 | |
| wt% | Avg | St.Dv. | Avg | St.Dv. | Avg | St.Dv. | Avg | St.Dv. |
| SiO$_2$ | 41.28 | 0.73 | 40.56 | 0.58 | 41.70 | 0.76 | 41.44 | 0.31 |
| FeO | 5.90 | 0.14 | 6.49 | 0.28 | 5.36 | 0.46 | 4.17 | 0.20 |
| MnO | 0.08 | 0.03 | 0.08 | 0.05 | 0.07 | 0.04 | 0.07 | 0.02 |
| MgO | 52.16 | 0.78 | 52.20 | 0.53 | 52.63 | 0.85 | 52.43 | 0.79 |
| NiO | 0.44 | 0.05 | 0.38 | 0.03 | 0.56 | 0.05 | 0.84 | 0.09 |
| CaO | 0.03 | 0.02 | 0.04 | 0.02 | 0.02 | 0.02 | 0.02 | 0.01 |
| ZnO | 0.03 | 0.04 | 0.01 | 0.02 | 0.01 | 0.02 | 0.02 | 0.03 |
| tot | 99.95 | 0.91 | 99.79 | 0.82 | 100.42 | 0.89 | 99.18 | 0.96 |
| Si | 0.99 | 0.02 | 0.98 | 0.01 | 1.00 | 0.02 | 1.00 | 0.01 |
| Fe$^{2+}$ | 0.12 | 0.00 | 0.13 | 0.01 | 0.11 | 0.01 | 0.08 | 0.00 |
| Mn | 0.00 | 0.00 | 0.00 | 0.00 | 0.00 | 0.00 | 0.00 | 0.00 |
| Mg | 1.87 | 0.02 | 1.88 | 0.01 | 1.88 | 0.02 | 1.89 | 0.01 |
| Ni | 0.01 | 0.00 | 0.01 | 0.00 | 0.01 | 0.00 | 0.02 | 0.00 |
| Ca | 0.00 | 0.00 | 0.00 | 0.00 | 0.00 | 0.00 | 0.00 | 0.00 |
| Zn | 0.00 | 0.00 | 0.00 | 0.00 | 0.00 | 0.00 | 0.00 | 0.00 |
| Mg# | 0.940 | 0.002 | 0.935 | 0.003 | 0.945 | 0.004 | 0.957 | 0.002 |

The MgO content of olivine in dunite ranges between 49.34 and 54.64 wt%, while the FeO content ranges between 5.56 and 6.90 wt%. Mg# varies between 0.930 and 0.943. The NiO content varies between 0.38 and 0.53 wt%.

The MgO content of olivine in massive chromitite varies between 50.34 and 53.80 wt%, and the FeO content ranges between 3.74 and 6.08 wt%. Mg# values are comprised between 0.939 and 0.962. The NiO content varies between 0.68 and 0.96 wt%.

Olivine and chromite Mg# values show a correlation with the distance from the grain boundary, both in dunites and chromitites (Figure 8).

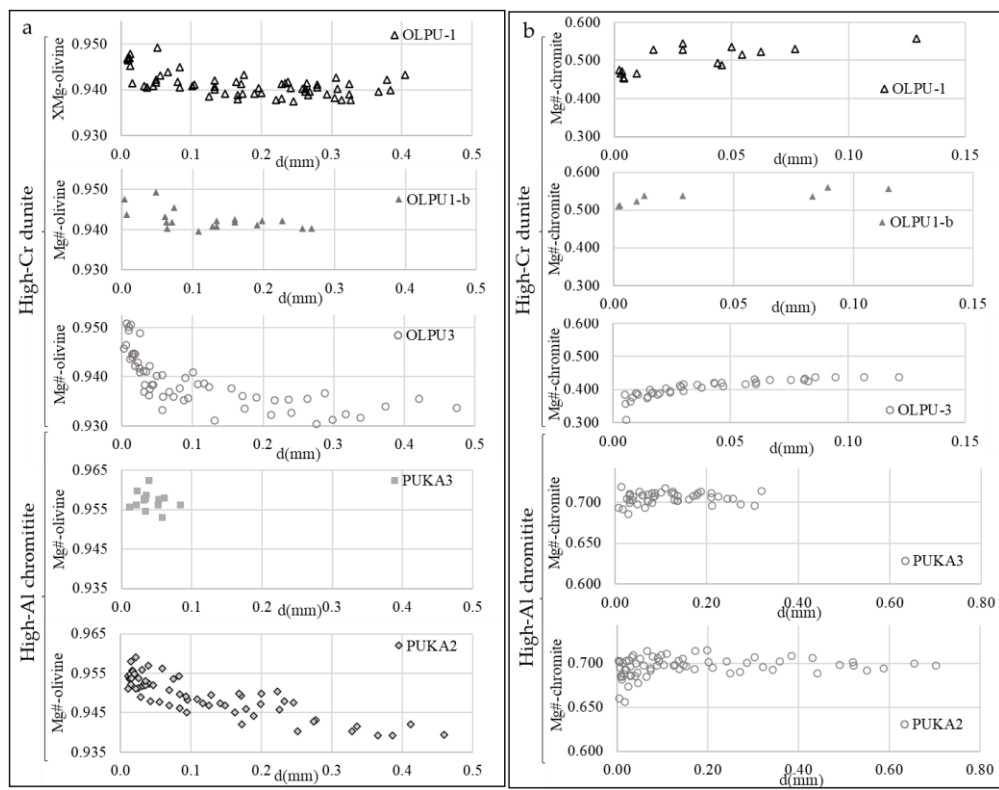

**Figure 8.** Mg# vs. distance (d) from the grain boundary (mm) of (**a**) olivines in dunites and chromitites and (**b**) chromites in dunites and chromitites at Iballe.

In dunites, olivine invariably shows an increase in Mg# up to 0.020 approaching the grain boundary. The zonation extends as far as 150 μm from the intergranular limit (Figure 8a). Chromite in dunites shows a zonation with a Mg# variation about 2 times larger than the one in olivines, and the extent of the zonation is narrower than 1000 μm (Figure 8b).

In chromitites, olivine shows a zonation only in sample PUKA-2, where the increase in Mg# close to the grain boundary is about 0.020, and the zonation extends as far as 350 μm (Figure 8a). It was not possible to detect any zonation within olivine of sample PUKA-3 due to the massive texture of the sample, as there were no points within olivines at a distance > 100 μm from the grain boundary.

Chromite in chromitites shows a decrease in Mg# close to the grain boundary. The decrease affects the grains up to 200 μm from the rim and the maximum Mg# variation is up to 0.05 (Figure 8b).

### 4.2.2. Nea Roda

Chromite average compositions (Table 3) are based on core analyses of fresh chromitite samples. Complete results of electron-microprobe analyses are reported in the Supplementary Materials. All analyzed samples are chromitites with a 40–60% spinel content, and a texture ranging from banded to nodular. The MgO content ranges between 5.28 and

10.22 wt%, with lower values in nodular chromitites (samples GR62 and GR63) and higher values in banded chromitite (sample GR58). The FeO content ranges between 18.05 and 25.69 wt%, and the $Fe_2O_3$ content is lower than 5.01 wt% for all chromitites. The $Cr_2O_3$ content ranges between 53.32 and 60.48 wt%, with higher values in banded chromitite than in nodular chromitites, and the $Al_2O_3$ content ranges between 8.84 and 13.83 wt%.

**Table 3.** Major and minor elements average composition and standard deviation of chromite cores from Nea Roda banded and nodular chromitites; Mg# = [Mg/(Mg + $Fe^{2+}$)]; Cr# = [Cr/(Cr + Al)].

| Lithology | Banded Chromitites | | Nodular Chromitites | | | |
| Sample | GR-58 | | GR-63 | | GR-62 | |
| wt% | Avg | St.Dev | Avg | St.Dev | Avg | St.Dev |
|---|---|---|---|---|---|---|
| $TiO_2$ | 0.10 | 0.01 | 0.21 | 0.06 | 0.17 | 0.03 |
| $Al_2O_3$ | 9.58 | 0.39 | 12.89 | 0.61 | 13.47 | 0.35 |
| $Cr_2O_3$ | 60.21 | 0.19 | 54.21 | 0.45 | 55.24 | 0.79 |
| $V_2O_3$ | 0.06 | 0.02 | 0.10 | 0.03 | bdl | bdl |
| $Fe_2O_3$ | 1.85 | 0.22 | 3.77 | 0.63 | 0.89 | 0.66 |
| FeO | 19.63 | 1.04 | 22.11 | 0.64 | 23.91 | 0.98 |
| MgO | 9.15 | 0.71 | 8.03 | 0.48 | 6.80 | 0.63 |
| tot | 100.91 | 0.28 | 101.75 | 0.39 | 100.83 | 0.54 |
| Ti | 0.00 | 0.00 | 0.28 | 0.26 | 0.00 | 0.00 |
| Al | 0.37 | 0.01 | 0.98 | 0.47 | 0.52 | 0.01 |
| Cr | 1.57 | 0.01 | 0.64 | 0.71 | 1.44 | 0.01 |
| V | 0.00 | 0.00 | 0.04 | 0.04 | 0.00 | 0.00 |
| $Fe^{3+}$ | 0.05 | 0.01 | 0.37 | 0.25 | 0.02 | 0.02 |
| $Fe^{2+}$ | 0.54 | 0.03 | 0.29 | 0.31 | 0.66 | 0.03 |
| Mg | 0.45 | 0.03 | 0.17 | 0.19 | 0.34 | 0.03 |
| Mg# | 0.808 | 0.007 | 0.732 | 0.003 | 0.733 | 0.006 |
| Cr# | 0.454 | 0.032 | 0.408 | 0.007 | 0.336 | 0.030 |

The slight differences in spinel mineral chemistry between chromitites with different textures are reflected in higher Mg# and Cr# of banded chromitites with respect to nodular chromitites.

Olivine mineral chemistry is based on core analyses of olivine crystals (Table 4). Olivine composition is strongly forsteritic and quite homogeneous in all samples. The MgO content varies between 50.1 and 55.0 wt% and the FeO ranges between 4.6 and 8.3 wt%. Mg# ratio varies between 0.92 and 0.95. Olivine also shows a Ni enrichment, varying between 0.32 and 0.54 wt%.

**Table 4.** Major and minor elements average composition and standard deviation of olivine cores from Nea Roda banded and nodular chromitites; Mg# = [Mg/(Mg + $Fe^{2+}$)].

| Lithology | Banded Chromitites | | Nodular Chromitites | | | |
| Sample | GR-58 | | GR-63 | | GR-62 | |
| wt% | Avg | St.Dev | Avg | St.Dev | Avg | St.Dev |
|---|---|---|---|---|---|---|
| $SiO_2$ | 41.19 | 0.28 | 39.63 | 0.98 | 42.29 | 0.41 |
| FeO | 4.81 | 0.18 | 6.70 | 0.15 | 7.30 | 0.35 |
| MnO | 0.08 | 0.02 | 0.10 | 0.03 | 0.10 | 0.06 |
| MgO | 51.86 | 0.19 | 51.28 | 0.85 | 51.43 | 0.51 |
| NiO | 0.43 | 0.05 | 0.43 | 0.04 | 0.37 | 0.03 |
| CaO | 0.01 | 0.01 | 0.01 | 0.01 | 0.01 | 0.01 |
| ZnO | 0.03 | 0.03 | 0.03 | 0.04 | 0.01 | 0.02 |
| tot | 98.49 | 0.38 | 98.30 | 0.87 | 101.60 | 0.62 |
| Si | 1.00 | 0.00 | 0.97 | 0.02 | 1.01 | 0.01 |
| $Fe^{2+}$ | 0.10 | 0.00 | 0.14 | 0.00 | 0.15 | 0.01 |
| Mn | 0.00 | 0.00 | 0.00 | 0.00 | 0.00 | 0.00 |
| Mg | 1.89 | 0.01 | 1.88 | 0.02 | 1.83 | 0.01 |
| Ni | 0.01 | 0.00 | 0.01 | 0.00 | 0.01 | 0.00 |
| Ca | 0.00 | 0.00 | 0.00 | 0.00 | 0.00 | 0.00 |
| Zn | 0.00 | 0.00 | 0.00 | 0.00 | 0.00 | 0.00 |
| Mg# | 0.951 | 0.002 | 0.932 | 0.001 | 0.926 | 0.003 |

At Nea Roda, olivine Mg# does not correlate with the distance from the grain boundary (Figure 9a). Olivine Mg# data are quite dispersed and do not present the increase in Mg# values close to the grain boundary typical of olivine-chromite re-equilibration. Chromite Mg#, on the other hand, decreases from the core to the rim of the crystals (Figure 9b). Chromite crystals display a zonation, with Mg# variation up to 0.20, extending as far as 0.3 mm from the grain boundary (Figure 9b).

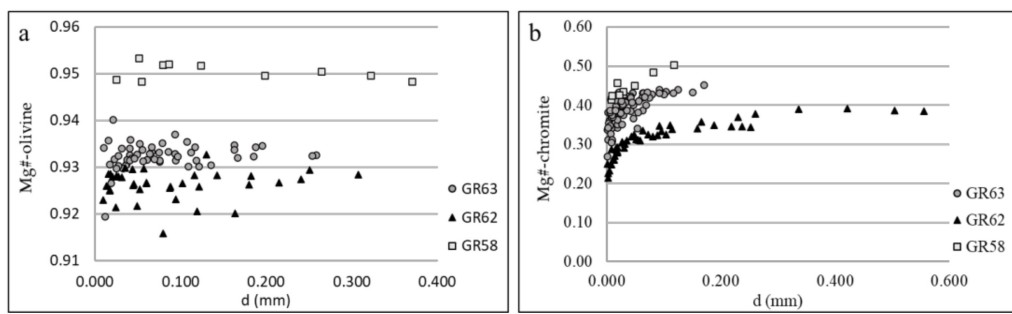

**Figure 9.** Mg# vs. distance from the grain boundary (mm) of (**a**) olivines and (**b**) chromites in banded and nodular chromitites at Nea Roda.

## 5. Discussion

### 5.1. Diffusivity Curve Modelling

Olivine and spinel compositions at grain boundaries reflect the temperature below which subsolidus exchange cannot proceed. On the contrary, at the core of grains, minerals are more likely to retain their primary composition, which can be used to estimate primary temperatures. As Mg# patterns follow an exponential trend, re-equilibrated values are very sensitive to the distance from the grain boundary. To minimize the potential error in the characterization of primary and re-equilibrated values, we used an exponential function (Equation (1)) to model Mg# diffusion patterns, following the methodology implemented by Bussolesi et al. [11]. The best fit exponential curve was calculated using the software OriginPro.

$$y = a - bc^x \tag{1}$$

where:

$y$ = Mg#

$x$ = distance (μm) from the grain boundary

$a$ = primary Mg# (for $x \rightarrow \infty$)

$(a - b)$ = re-equilibrated Mg# (for $x \rightarrow 0$)

$b$ = primary Mg# − re-equilibrated Mg#

$c = \frac{(a-y)}{b}$ (for $x = 1$ μm) = $\frac{(\text{primary Mg\#} - \text{Mg\# for } x = 1 \text{ μm})}{(\text{primary Mg\#} - \text{re-equilibrated Mg\#})}$ = normalizing parameter depending on the unit of measurement chosen for distance ($x$).

Diffusivity curves at Iballe were modeled for spinel and olivine datasets (Table 5, Figure 10). The computation of primary and re-equilibrated Mg# values of olivine was always possible for dunite, whereas it was only possible for one chromitite sample, which shows a sufficiently high *r* factor (olivine/(olivine + chromite). When the *r* factor is too low, olivine is more likely to be completely re-equilibrated with surrounding chromite and does not show Mg# variations from core to rim. In one of the Iballe chromitites, olivine crystals have a small grain size which prevented them from retaining the primary composition at the core. In that case, an average Mg# was used for geothermometry calculations.

**Table 5.** Parameters a, b, c and associated standard errors for the calculation of diffusivity curves and primary (pr) and re-equilibrated (eq) Mg# of chromite and olivines in Iballe dunites and chromitites.

| Sample | Rock | a | St.Er. | b | St.Er. | c | St.Er. | Mg#$_{pr}$ | Mg#$_{eq}$ |
|--------|------|---|--------|---|--------|---|--------|------------|------------|
| | | | | | Chromite | | | | |
| OLPU-1 | Dunite | 0.527 | 0.010 | 0.081 | 0.018 | 0.924 | 0.052 | 0.527 | 0.446 |
| OLPU1-b | Dunite | 0.551 | 0.006 | 0.044 | 0.010 | 0.941 | 0.034 | 0.551 | 0.507 |
| OLPU-3 | Dunite | 0.440 | 0.006 | 0.080 | 0.005 | 0.974 | 0.005 | 0.440 | 0.360 |
| PUKA2 | Chromitite | 0.700 | 0.002 | 0.014 | 0.005 | 0.975 | 0.018 | 0.700 | 0.685 |
| PUKA3 | Chromitite | 0.707 | 0.001 | 0.010 | 0.008 | 0.965 | 0.038 | 0.707 | 0.697 |
| | | | | | Olivine | | | | |
| OLPU-1 | Dunite | 0.940 | 0.000 | −0.007 | 0.001 | 0.979 | 0.007 | 0.940 | 0.948 |
| OLPU1-b | Dunite | 0.941 | 0.001 | −0.006 | 0.002 | 0.987 | 0.010 | 0.941 | 0.947 |
| OLPU-3 | Dunite | 0.934 | 0.001 | −0.016 | 0.001 | 0.973 | 0.005 | 0.934 | 0.951 |
| PUKA2 | Chromitite | 0.934 | 0.007 | −0.020 | 0.006 | 0.997 | 0.002 | 0.934 | 0.955 |
| PUKA3 | Chromitite | n.c. | n.c. | n.c. | n.c. | n.c. | n.c. | 0.957 * | 0.957 * |

\* Calculated as average Mg# value; n.c. not computable.

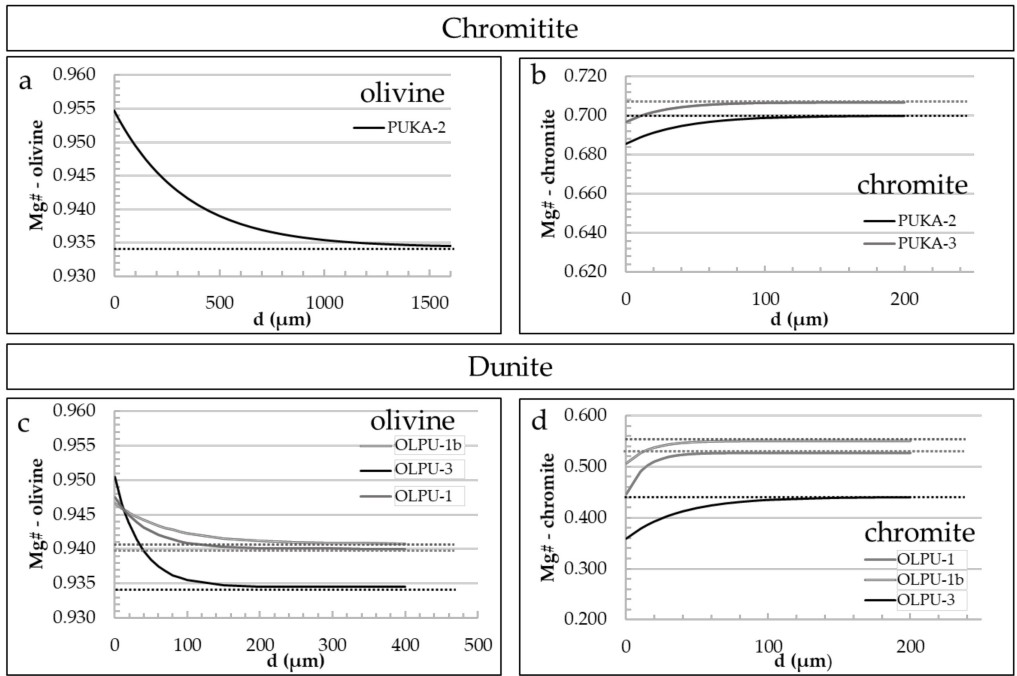

**Figure 10.** Diffusivity curves determined for olivine and chromite in dunites and chromitites of Iballe; the curves were computed through OriginPro software. (**a**) diffusivity curves of olivine in chromitites, (**b**) diffusivity curves of chromite in chromitites, (**c**) diffusivity curves of olivine in dunites, (**d**) diffusivity curves of chromite in dunites.

Chromite Mg#, on the contrary, was modeled for all the analyzed samples.

The standard error values associated to the parameters of the equation (Table 5) reveal that most of the inferred parameters are reliable. Primary Mg# (Mg#pr) has a relatively low standard error, as it is associated with parameter "a", whereas re-equilibrated Mg# (Mg#eq) has a higher error, as the parameter "b" has a higher standard error.

Diffusion patterns of olivine and chromite crystals are different in the two lithologies. In massive chromitites, the only olivine pattern has a diffusion distance comprised between 500 and 1000 μm (Figure 10a). Olivine patterns within dunites, on the other hand, have a diffusion distance comprised between 80 and 150 μm (Figure 10c).

Chromite diffusion patterns extend at a lower distance than olivine ones. The diffusion distance ranges between 40 and 50 μm from the grain boundary within chromitites (Figure 10b), and between 20 and 100 μm within dunites (Figure 10d).

At Nea Roda, none of the olivines shows a trend in Mg# from core to rim, so it was not possible to model diffusion patterns (Figure 9). The lack of diffusion patterns in olivine, despite the fact that the *r* factor is not low enough for olivine to re-equilibrate with the surrounding chromite, is at odds with the Mg# variability in Nea Roda chromites. However, especially in ophiolite environments, spinels can be subject to partial transformation into ferrian chromite which can produce an effect similar to the one of re-equilibration with olivine. At Nea Roda ferrian chromite is quite widespread, much more than at Iballe (Figure 7a), and the patterns in chromite are likely caused by this spinel alteration.

*5.2. Geothermometry*

At Iballe, temperature estimates were applied to re-equilibrated and primary Mg# values according to the Ballhaus et al. [6] geothermometer (Table 6). The olivine-spinel re-equilibrated temperatures are 677 and 803 °C within chromitites, and range between 527 and 724 °C in dunites. Primary temperatures for chromitites are 821 and 996 °C. Temperatures recorded for dunites are lower, ranging between 655 and 806 °C.

**Table 6.** Primary and re-equilibrated temperatures (Tpr and Teq) of Iballe chromitites and dunites.

| Sample | Lithology | Tpr (°C) | Teq (°C) |
|--------|-----------|----------|----------|
| OLPU-1 | Dunite | 778 | 659 |
| OLPU-1b | Dunite | 807 | 724 |
| OLPU-3 | Dunite | 655 | 527 |
| PUKA2 | Chromitite | 998 | 677 |
| PUKA3 | Chromitite | 822 * | 805 * |

* Values are based on averages and are underestimated (Tpr) and overestimated (Teq).

Re-equilibration temperatures record the closure temperatures of the subsolidus exchange. Within peridotites, olivine-spinel is the system that "freezes" last at decreasing temperatures, allowing the determination of the thermal history of the rocks down to ~650 °C [8,10,31]. The primary temperature represents the temperature below which diffusivity cannot maintain compositional homogeneity within the crystals [2,8,31,32].

Iballe temperatures are consistent with geothermometry data obtained in previous works for mafic-ultramafic cumulates of the Eastern and Western Mirdita Ophiolites [18], which range between 725 and 810 °C.

At NeaRoda, the absence of diffusion patterns in olivine crystals allows only the calculation of a primary temperature using core analyses. In Table 7 the average core values and temperature estimates are reported using the Ballhaus et al. [6] geothermometer. Estimated temperatures for banded and nodular chromitites vary between 550 °C and 656 °C. These are in agreement with Nea Roda olivine-spinel thermometry reported in the literature [28], yielding temperature estimates comprised between 530 and 620 °C according to the Roeder et al. calibration [33], and between 640 and 745 °C according to the Fabriès calibration [8]. These low temperatures have been interpreted as re-equilibration temperatures [28].

**Table 7.** Primary temperatures at Nea Roda, calculated using average Mg# at the core of the minerals.

| Sample | Lithology | Tpr (°C) |
|--------|-----------|----------|
| GR58 | Banded chromitite | 633 |
| GR63 | Nodular chromitite | 656 |
| GR62 | Nodular chromitite | 551 |

Completely re-equilibrated patterns in large crystals (millimetric diffusion distances) are not compatible with postmagmatic re-equilibration. They can be better explained with a long-lasting metamorphic peak, also compatible with the relatively low temperatures calculated, not fitting the cooling of the magmatic system.

### 5.3. Cooling Rates

Mg-Fe$^{2+}$ zoning used to infer primary and re-equilibrated compositions and temperatures can also be applied to the estimate of a cooling rate in ultramafic rocks [34,35]. Ozawa [34] calculated cooling rate profiles for several types of ultramafic rocks, at different initial temperatures of the system.

It was possible to assess cooling rates only for Iballe, as Nea Roda does not have diffusivity patterns. Cooling rates were assessed through comparison with cooling rate profiles calculated by Ozawa [35] for the Iwanaidake peridotites. Constant cooling rate curves chosen for comparison are calculated for Cr# = 0.50 (Iballe massive chromitites) and Cr# = 0.78 (Iballe dunites) and were plotted in a semi-logarithmic chart reporting grain diameter vs. T (°C). Iballe diffusion curves have been calculated for one chromitite sample, and for an average between dunite samples. Of the two chromitite samples, one was not considered due to lack of diffusion patterns in olivine. Iballe diffusion curves were redrawn assuming that the diameter of spinel grains, is equal to two times the distance from the grain boundary (d) (as considered by Ozawa [35]).

Diffusion curves (Figure 11) are characterized by a flat part at higher distances representing the primary temperatures at mineral cores, and by a steep part in the re-equilibration zone. The steep portion describes the cooling rate of the samples. The curves can either follow a constant cooling rate or, more often, deviate from it, implying a variable cooling rate.

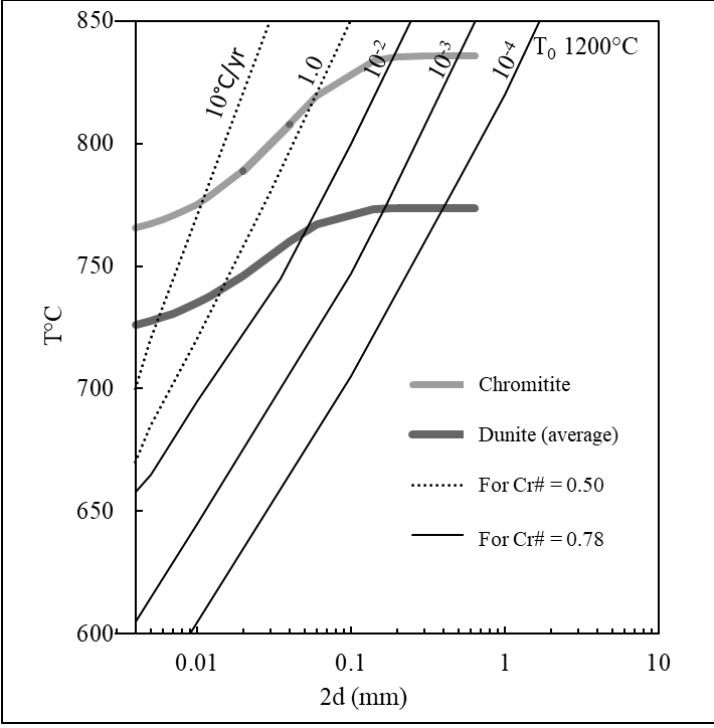

**Figure 11.** T (°C) vs. 2D plot of Iballe diffusion curves. Temperatures are calculated according to Fabriès [8]. Constant cooling rate curves for Cr# = 0.50 and Cr# = 0.78 are from Ozawa [35].

Results show that Iballe cooling rates are different for dunites and chromitites. The former show lower temperatures (Table 6) and a cooling rate variable but always higher than $10^{-2}$ °C/yr.

Massive chromitites show steeper curves, implying an almost constant cooling rate, comprised between 1 and 10 °C/yr. These different rates can be explained with a different genetic history of chromitites and dunites, as suggested also by the differences in spinel mineral chemistry.

### 5.4. Iballe Genetic History

Chromitites in ophiolites are divided into high-Cr chromitites, which are widespread and generated from boninitic magmas in suprasubduction settings, and high-Al chromitites, generated from less refractory magmas either at mid-ocean ridge or in suprasubduction settings [36–38]. Numerous cases of ophiolite massifs containing both types of chromitites have been reported [39–42], with high-Cr chromitites stratigraphically located in the mantle and high-Al chromitites within the Moho Transition Zone, which is a level between the lithospheric mantle and the overlying oceanic crust [43–45].

Co-existing high-Al and high-Cr chromitites are generally explained by the onset of a subduction zone (hosting high-Cr chromitites) close to a back-arc region (hosting high-Al chromitites) [18,46,47]. Chromitites formed in different geodynamic settings can sometimes occur in tectonic contact after being deformed and displaced during obduction [48].

Qiu et al. [41] analyzed spinels from high-Al and high-Cr chromitites, mantle dunites and MTZ dunites from the Bulqiza ultramafic massif, located to the South of Mirdita ophiolite in central Albania. Spinels from high-Al chromitites and associated MTZ dunites show Cr# lower than 0.6, while spinels from high-Cr chromitites and associated mantle dunites have Cr# higher than 0.8. Iballe spinels are chemically comparable to Bulqiza high-Al chromitites and mantle dunites (Figure 12).

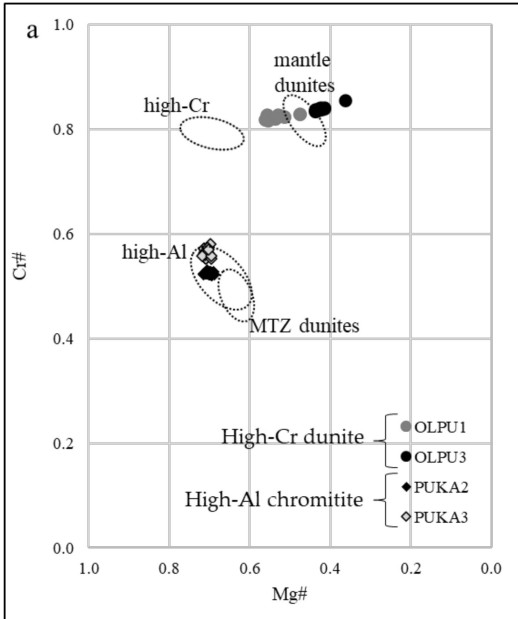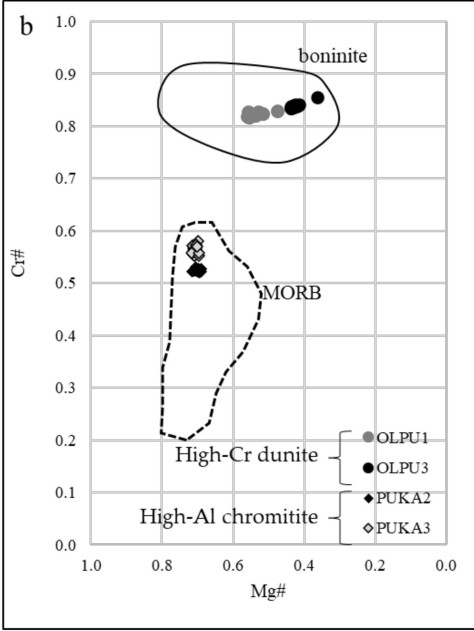

**Figure 12.** Cr# vs. Mg# of Iballe high-Al chromitites and high-Cr dunites compared to (**a**) dunites and chromitites of the Bulqiza massif of Northern Albania [41]; (**b**) boninites and MORB compositional fields from Barnes and Roeder [49].

From spinel mineral chemistry we can thus infer that Iballe dunites and chromitites analyzed in the present contribution are not genetically related. Chromitites contain high-Al spinels with compositions similar to high-Al chromitites from Bulquiza. Dunites host scattered Cr-spinel whose composition matches the high-Cr array of Bulquiza chromitites and dunites. High-Cr spinels in Iballe dunites show a boninitic affinity (Figure 13), whereas

chromitites have high-Al spinels similar to MORB ones, but with minor differences (e.g., slightly higher Cr# and slightly lower $TiO_2$).

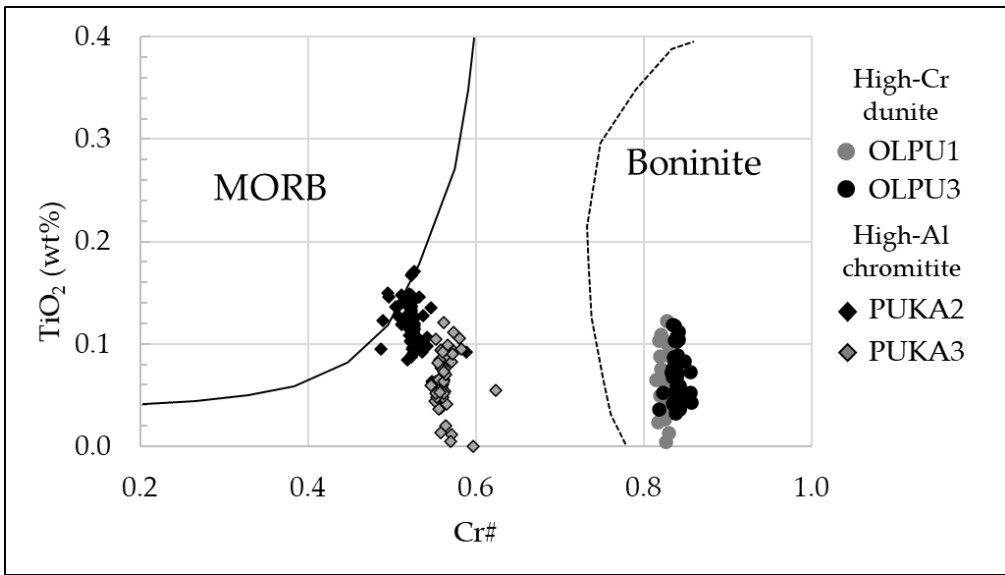

**Figure 13.** The $TiO_2$ content of spinel vs. its Cr# value in Iballe high-Cr dunites and high-Al chromitites; compositional fields of MORB and boninite are from Dick and Bullen [50].

However, Iballe high-Al chromitites have a $TiO_2$ content lower than typical MORB spinels (Figure 13), implying that these chromitites did not form directly from MORB magmas, but from the partial melting of a residual mantle which had already experienced a low degree of melt extraction [41], typical of SSZ settings.

Saccani and Tassinari [18] suggested that the low-Ti and high-Al contents of spinels in the Krabbi massif, where Iballe is located, might have originated from low-Ti tholeiites. This model explains the coexistence of MORB and SSZ-type melts through the establishment of a subduction zone close to an active mid-ocean ridge, at high thermal regimes. In the model, the Western Mirdita Ophiolites are located very close to the subduction zone, and the proximity of both MORB and SSZ-type sources allowed minor, late MORB-type melts to be generated at the same time as boninitic melts in the Eastern Mirdita Ophiolites [18,51,52].

However, recent studies suggest the possibility that the wide chromite compositional range may derive from an evolving mantle source from MORB-like to boninitic during the initial stages of subduction, in a proto-forearc setting [41,53–55].

At subduction initiation, melts generated by decompressional melting in the extensional region above the subducting plate [56,57] have a MORB-like composition, and can yield high-Al chromitites. During the evolution of the subduction zone, the parental melt changes composition due to a major release of hydrous fluids from the subducting slab, which creates the optimal condition for the formation of high-Cr chromitites and their associated dunites [54,58,59] (Figure 14).

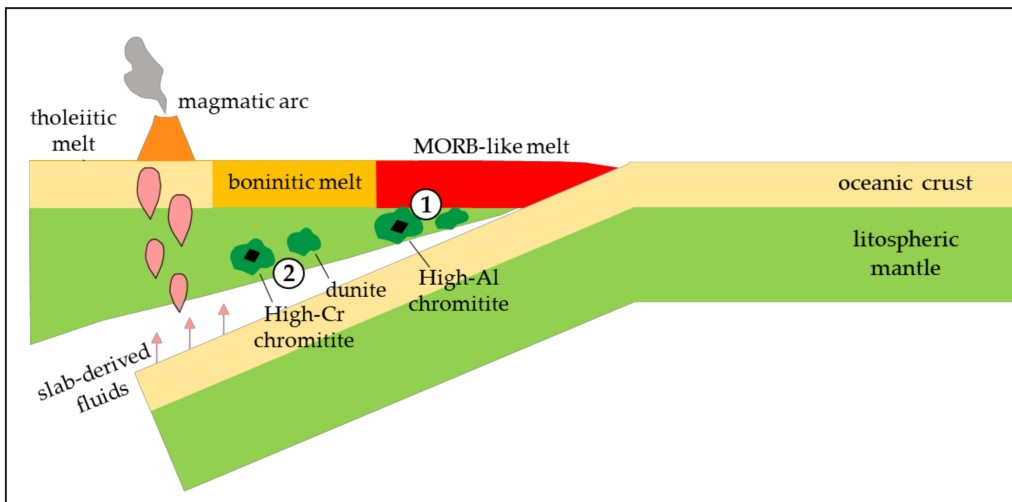

**Figure 14.** Schematic diagram showing the formation of Iballe chromitites and dunites in a proto forearc setting, adapted from the Chen et al. [55] model; (1) melts generated through decompressional melting at the onset of subduction have a MORB-like composition and can produce high-Al chromitites and dunites bearing high-Al spinel. (2) With the evolution of the subduction zone, slab-derived fluids generate boninitic melts together with dunites bearing high-Cr spinel and possibly high-Cr chromitites.

### 5.5. Nea Roda Genetic History

The genesis of Iballe high-Al chromitites in a proto-forearc setting is consistent with the inferred temperatures and cooling rates, between 1 and 10 °C/yr. Iballe high-Cr spinel-bearing dunites are related to a later stage of subduction, when boninitic melts are produced. Their lower cooling rate, around $10^{-2}$ °C/yr, may be due to their deeper position. These dunites could potentially host high-Cr chromitites, similar to those at Bulquiza, but those do not outcrop at Iballe and/or are not yet known.

Nea Roda ophiolite spinels are classified in the literature based on their host lithology in [28]: disseminated chromitites, dunites, harzburgites and massive chromitites (Figure 15). Following this classification, our samples could be classified as disseminated chromitites and dunites. However, the specimens analyzed for the present study are all chromitites, and will therefore be classified in the present contribution based on their texture as banded and nodular chromitites.

Spinels from both lithologies show clues of transformation into ferrian chromite such as pores on chromite surface and light gray rims. The alteration of chromite into ferrian chromite can create a "fake" diffusion pattern, in which the decrease in Mg content is not a consequence of re-equilibration with the surrounding olivine but of a loss of Cr and relative increase in Fe. As olivine is not affected by alteration to ferrian chromite, the absence of diffusion patterns in olivine allows the discrimination of fake diffusion patterns in chromite, such as those found in Nea Roda chromitites.

The absence of diffusion patterns can be interpreted as a complete obliteration of all the primary features by a relatively high-T post-genetical process. Re-heating events can completely cover diffusion patterns if a high enough temperature is maintained for a long period of time [11,28]. The primary temperature estimated through core analyses would be in this case a re-equilibration temperature.

Michailidis et al. [28] reported that Nea Roda ultramafic rocks display low P-T greenschist facies metamorphism. However, recorded temperatures are between 550 and 650 °C, thus suggesting a higher metamorphic grade, probably related to a Permo-Triassic amphibolite-facies event (T = 500–640 °C, P = 5–8 kbar) [28,60].

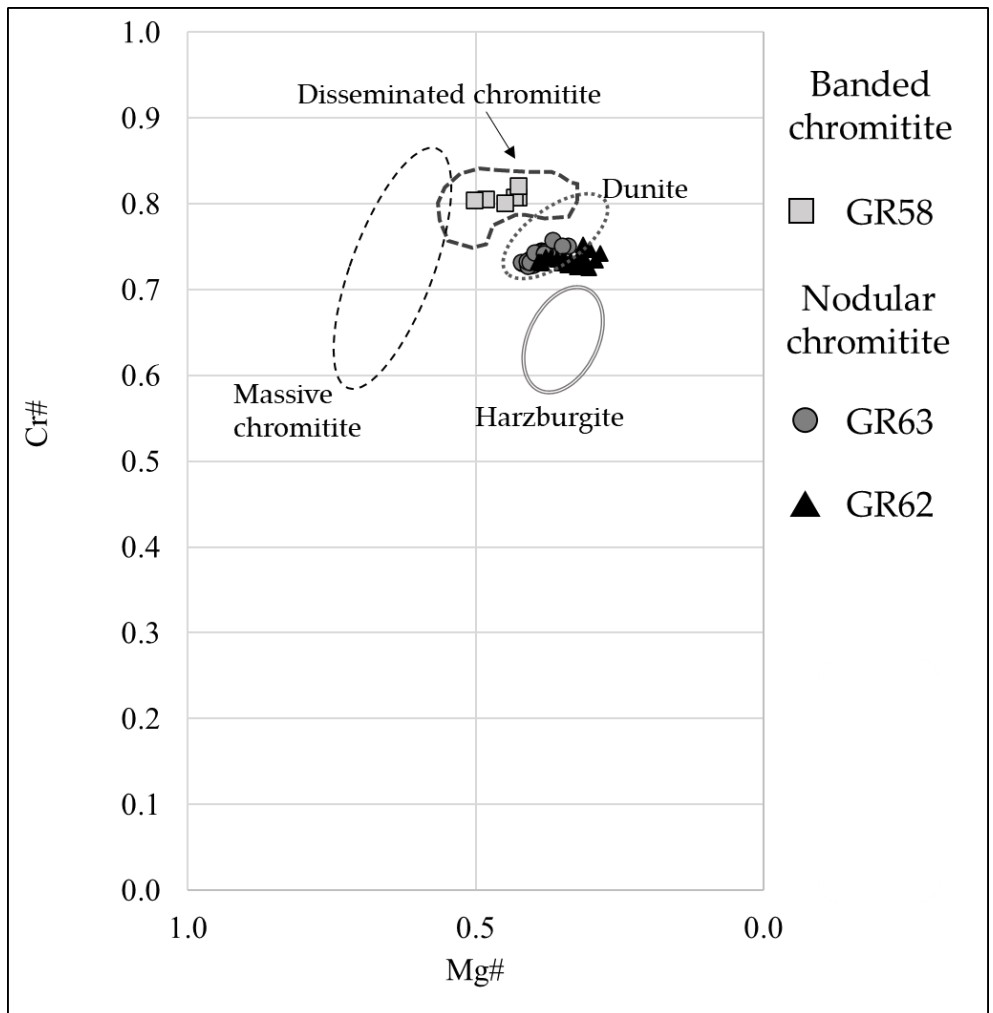

**Figure 15.** Cr# vs. Mg# of Nea Roda spinels; compositional fields of massive chromitite, harzburgite, dunite and disseminated chromitite are from Michailidis et al. [28].

The complete re-equilibration of olivine and spinel has to be partially held responsible for the high FeO content of chromites. A change in chemical composition of chromite cores prevents us from deducing the genetic history of the Nea Roda massif.

## 6. Conclusions

Two sets of samples were collected at the Iballe massif: high-Cr dunites and high-Al chromitites. Geothermometry and geospeedometry assessments of the two sets of samples reveal that chromitites were formed at relatively high temperatures and suffered a rapid cooling at nearly constant cooling rate. Dunites reveal lower temperatures of formation and lower cooling rates than chromitites. This remarkable difference in cooling and genetic conditions is also reflected in the spinel chemistry. Chromitite spinels have low Cr and high Al contents, quite similar to MORB spinels, whereas dunite spinels are high-Cr, and show a boninitic affinity. High-Al chromitites with MORB affinity formed at an early stage of proto-forearc in a SSZ setting, explaining the higher cooling rates, while high-Cr dunites with boninitic affinity were formed deeper in the mantle in a more mature subduction setting.

The Nea Roda ophiolite hosts small disseminated chromitite bodies enclosed within dunite dykes. Mineral chemistry analyses of spinel and olivine along traverses highlighted the absence of diffusivity patterns in olivines, and the presence of "fake" patterns within spinels, probably due to transformation into ferrian chromite. This allowed only the calculation of a temperature using core data. Resulting temperatures are between 550 °C to 656 °C

in agreement with previous studies on the same area. These low temperatures, coupled with the absence of diffusion patterns, can be explained by a complete re-equilibration during a relatively high-T metamorphic event, causing the complete obliteration of any primary feature.

**Supplementary Materials:** The following are available online at https://www.mdpi.com/article/10.3390/min12010064/s1, Table S1: complete results of electron-microprobe analyses of chromite and olivine crystals in chromitites and dunites of Iballe and Nea Roda ophiolites.

**Author Contributions:** Conceptualization, M.B. and G.G.; methodology, G.G. and F.Z.; software, M.B.; validation, A.C., G.G. and F.Z.; formal analysis, M.B. and G.G.; investigation, M.B.; resources, G.G. and F.Z.; data curation, M.B.; writing—original draft preparation, M.B.; writing—review and editing, G.G.; visualization, A.C.; supervision, F.Z. All authors have read and agreed to the published version of the manuscript.

**Funding:** This research was funded by the Italian Ministry of Education (MUR) through the project "PRIN2017—Mineral reactivity, a key to understand large-scale processes and the project 'Dipartimenti di Eccellenza 2017'".

**Data Availability Statement:** All data are available in the article and in the Supplementary Materials.

**Acknowledgments:** Authors acknowledge Shpetim Kastrati for the help during field work in Albania. We also wish to thank the reviewers that helped us improve the original manuscript with their valuable comments.

**Conflicts of Interest:** The authors declare no conflict of interest.

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
