# Peer review of "Different Tectonic Evolution of Fast Cooling Ophiolite Mantles Recorded by Olivine-Spinel Geothermometry: Case Studies from Iballe (Albania) and Nea Roda (Greece)"

_minerals, doi:10.3390/min12010064_

Round 1

Reviewer 1 Report

This is an interesting paper on the origin of chromitites in ophiolitic peridotites.

For me, as I am a reader ignorant of the local geology, the cornerstone question for the discussion: how can dunites possibly be unrelated to chromitites? One of the key properties of the modern conception is that chromitites formed as a result of a melt-rock interaction followed by incongruent orthopyroxene melting, formation of boninitic melt, magma mixing etc. Moreover, it is unclear how can chromitites be older than dunites if “Dunites are mostly fresh, and host small chromitite lenses up to 10 cm thick” (line 147-148)?                                                                                

Also, there is another problem. Authors state that massive chromitites have had a cooling rate of 1­ to 10 ºC/yr, while dunites have cooling rates of ≈ 0.01 ºC/yr. Apparently, the difference is of 2-3 orders of magnitude! Authors should explicitly suggest what geological processes can be responsible for such differences. The problem is that, according to the author's ideas, chromitites in Iballe were formed before dunites (which, once again, sound strange to me).  Hence, chromitites must have continued cooling together with dunites at the slow rates of the latter. Hence, the curves for chromitites must have been affected by this slow cooling. I am not trying to say that there is any critical problem, I just want to emphasize that the paper lacks explicit explanations of some genetic points.

The possible answer to this question can be found on lines 402-405, where the authors mention that chromtities formed in different settings may occur in tectonic contact. However, it is still unclear to what extent can this idea be applied to the studied samples. For instance, lines 142-144 state that chromtities occur in small lenses in dunites. Do I understand it correctly that high-Al chromitite veins (older) occur in the dunites of boninite origin (younger)? I believe that no, but you must make it clear.

Hence, I want to emphasize that the whole question is not about scrutiny, but about explanation. I believe that the authors have a sufficient model in their minds, and it is just a matter of a few sentences and, possibly, one figure to express them explicitly. Please, consider addressing the following:

1) Tell us more about the location of the samples. Could you expect that chromtities and dunites of the Ibale have different origins on a basis of geological/textural relations (before you conducted the study)?

2) Draw a small scheme summarizing the "Iballe Genetic History".

The second problem I would like to address is the Tpr - temperature of Cr-spinel crystallization according to the Ol-Spl thermometry. Apparently, the values for dunites (778, 807, 655) are well below any reasonable magmatic temperatures. Similar is for chromitites. While their formation temperatures are higher, they are still too low for high-Mg melts that, as many researchers believe, are parental for chromitites (e.g. Rollinson et al., 2018). I personally believe that the problem of chromitites origin is very far from being solved, and you can use this opportunity to speculate on their nature. Was it due to the hydrous nature of fluids? Any subsolidus processes involved? Anyway, at least a provisional explanation must be provided for the reported temperatures.

Minor corrections:

Line 68. Consider deleting "belong".

Line 88. Check in the word "Authors" should be capitalized.

Figure 1. Please, add an inset with the global position of Miridita ophiolite, as it is done for Nea Roda on Fig. 2.

Lines 293-296. Please, indicate what does "c" stands for in the formula.

Line 394. The previous section was 5.2, so this section must be 5.3 instead of 5.1.

Line 449. Same as previous.

Reviewer 2 Report

In this contribution, the authors present valuable geothermometric information about chromite or magnesiochromite that has equilibrated with magnesian olivine in two ophiolitic suites, one in Albania, one in Greece. At the beginning of my review, I was impressed with the clarity of expression of the authors. As I continued, however, I became totally distracted by the linguistic problems of the authors. My main recommendation to them: find someone who is fluent in English and who knows about chromium-bearing spinel. Your text is not ready for publication. I have made a large number of suggested corrections, but it is the responsibility of the referee to submit a text that has been corrected, and in which the quality of English writing is at least equal to the quality of the scientific content of the article. Please refrain from using terminology like "Fe-chromitization". The verb "to chromitize" does not exist, and in Italian either! Basic improvements are needed in the construction of a paragraph; the names of minerals should be singular!Why did I not encounter the word magnesiochromite in this article? I hope that you will succeed in getting help to finalize your text.

As a scientific question, I would have liked to read about just what are the diffusing species in the creation of a diffusion profile like you show. Are they only cations or is H2O somehow involved? In other words, is it really diffusion that is going on, or is it some complex solution-and-redeposition reaction? But this is perhaps the topic of a future article...

Round 2

Reviewer 1 Report

Dear Authors and Editors,

I believe that the paper may be considered for publication in its present state.

Author Response

No changes required by reviewer.

Reviewer 2 Report

Line 120:  I prefer to use these terms the way they were defined, ultramafic pertains to minerals, ultrabasic pertains to rocks. The fact that the authors you quote don't agree is not a reason not to modify your text.

Line 206: alteration to, not into (several other places in the text also)

Line 240:  always in Italian is sempre. I prefer to use "invariably"

Line 320: "as" is correct. Since in Italian is "da allora"

Author Response

We accepted all suggested revisions.